# Study on the Pseudo-Slope Length Effect of Buried Pipe Extraction in Fully Mechanized Caving Area on Gas Migration Law in Goaf

Pengxiang Zhao [1,2,3,4], Xingbao An [1,2,*], Shugang Li [1,2,3], Xinpeng Kang [1,2], Yitong Huang [5], Junsheng Yang [5] and Shikui Jin [4,6]

1 College of Safety Science and Engineering, Xi'an University of Science and Technology, Xi'an 710054, China
2 Key Laboratory of Western Mine Exploitation and Hazard Prevention Ministry of Education, Xi'an University of Science and Technology, Xi'an 710054, China
3 Western Engineering Research Center of Mine Gas Intelligent Drainage for Coal Industry, Xi'an 710054, China
4 Xinjiang Uygur Autonomous Region Coal Science Research Institute, Urumqi 830000, China
5 Liuhuanggou Coal Mine, Yankuang Xinjiang Mining Co., Ltd., Changji 831100, China
6 Xinjiang Coal and CBM Engineering Technology Research Center, Urumqi 830000, China
* Correspondence: 20220226069@stu.xust.edu.cn

**Abstract:** To study the law of gas transportation in mining areas, Fluent numerical simulation software was applied to examine the influence of different pseudo-slope lengths (PSL) on gas concentration in a U-ventilated working area under no-extraction conditions. Based on this, numerical simulation experiments were conducted on the buried pipe extraction arrangement parameters. The simulation found that when there was no extraction, the PSL had an impact on the airflow in the extraction area, which caused the airflow in the extraction area to be disordered, causing gas to accumulate locally at the working area. When the buried pipe depths (BPDs) and PSLs of the working area worked together, the gas concentration of the working area was lower when the inlet air influence zone and the extraction influence zone were through; otherwise, gas concentration accumulation occurred at the working area. The research results showed that when the PSL was at 25 m and BPD was at 20 m, the gas concentration at the working area was not abnormal, and the gas concentration in the upper corner was lower. By adjusting the PSL and BPD of the test working area, the maximum gas concentration in the upper corner was reduced to 0.46% and the maximum gas concentration in the return air outlet was reduced to 0.41%. The experimental and practical results provide important reference values for coal and gas co-mining.

**Keywords:** buried pipe extraction; inclined thick coal seam; pseudo-slope length; u-shaped ventilation; upper-corner gas concentration





## 1. Introduction

China is one of the few countries in the world where coal is the main energy source. Currently, the coal mined underground accounts for 81.8% of the total coal [1,2]. Coal seams generally have high gas content and low permeability. The rapid increase in frequent mining disturbances and unloading gas emission intensity is caused by the continuous increase in the mining intensity. This led to frequent gas overrun problems in the return airway and upper corner. It hindered the safety production of the mine [3–5]. The concentration of gas extracted during coal mine production is less than 30%. This value is lower than the national standard for gas use. Many coal mines still adopt direct emptying treatment, resulting in more than 20 billion cubic meters of gas being discharged into the atmosphere every year during coal mining operations. The amount of gas pollution discharged into the atmosphere is tens of times more than that of $CO_2$. At present, the rapid development of clean coalbed methane resources to replace the demand for coal energy has become increasingly urgent. After more than ten years of development, the theory of coal and gas

co-mining has achieved a more comprehensive technical standard. However, the geological environment of the coal seams in China is complex. The existing theoretical technology still cannot support the safe and efficient co-mining of coal and coalbed methane.

To ensure safe and efficient production in mines, scholars had established different experimental models for various factors inside the mining area based on different research purposes, and have researched the evolution of laws related to wind volume, temperature field, and pressure field [6–9], etc. Using computational fluid dynamics simulation, Chang et al. [10] studied the amount of air supply to the working area and the negative ventilation pressure used for gas discharge in the roof channel to obtain the most suitable ventilation pressure. Shao et al. [11] simulated the gas control effect of "U + L" and "Y + L" ventilation modes and optimized the main ventilation factors of the "U + L" ventilation mode by orthogonal test and fuzzy evaluation method. The results showed that the gas concentration of the "U + L" ventilation mode was lower than that of the "Y + L" mode. The optimized "U + L" ventilation mode effectively reduced the gas concentration in the upper corner and returned it to the air tunnel. Yang et al. [12] investigated the effect of the gas concentration distribution pattern of the "Y + HLDR" ventilation method under different air supply conditions with the help of COMSOL. The results showed that with an increase in the air supply, the gas concentration near the working area decreased, and the high-concentration gas was directed to the deeper part of the mining area. Li et al. [13] simulated the gas concentration and flow field under different wind speeds by constructing a cavity collapse rock model in the mining area. Liang et al. [14] built a two-dimensional distribution model of permeability in the mining area based on different collapse patterns. Wang et al. [15] combined FLAC3D and Fluent simulation methods to obtain the flow field of air leakage in the mining area. Wang et al. [16] studied the flow field distribution and air leakage in the extraction zone under different conditions for the long-walled working area. As for field tests, Zhu et al. [17] qualitatively analyzed the non-synchronous correlation characteristics of gas transport in the direction of airflow at the working area and proposed an algorithm for identifying gas monitoring data anomalies based on spatiotemporal correlation analysis. Yu et al. [18] studied the asymmetry of the flow field in the mining area. The location of the gas aggregation zone varies with the ventilation parameters, which can be used as an evaluation index to study the degree of air leakage in the mining area. Zhang et al. [19] analyzed the development height of fracture zone in an inclined coal seam by using FLAC and used Fluent simulation to optimize the arrangement parameters of a high-level directional long borehole, instead of a high extraction lane, for efficient extraction. Brigida et al. [20] processed the dynamic data of gas concentration by using an optimization algorithm to obtain the cyclic nonlinear distribution of mine gas subject to perturbation. Dzhioeva et al. [21] studied the relationship between mine longwall distance and gas as a function of distance and used polynomial regression to analyze the results in order to improve the prediction reliability of the dynamic distribution of gas.

Concerning the problem that gas concentration in the upper corner of high gas mines easily exceeds its limit, many scholars have analyzed the possible causes from different angles and have carried out field verifications [22,23]. Li et al. [24] established the physical model of the mining area and have proposed a combined drainage technology of buried and sprinkled pipes in the upper mine corner, proving the effectiveness of the technology through Fluent simulation experiments. Chen et al. [25] studied the effects of air intake and advancement speed on the gas concentration in the upper corner and obtained the best airspeed of 2 m/s for the air intake lane and the best speed of 3 m/d for the working area mining. Liu et al. [26] compared the gas pressure distribution and methane concentration distribution under the conditions of no extraction measures, buried pipe extraction in the upper corner, and gas extraction holes at different locations with the help of numerical simulations. The results showed that the gas extraction hole can be used to discharge a large amount of gas and can effectively control gas transport. Chai [27] established different arrangement parameters of high extraction lanes for the problem of gas over-limit in the upper corner during the rapid advance of the extra-thick coal seam. In addition, based

on a field verification, we realized the safety requirements of gas control at the working area. Xiong et al. [28] proposed secondary extraction under double stress interference and divided the coal body of the working area into three zones based on extraction concentration: high efficiency zone, effective zone, and original extraction zone. Among them, the average extraction concentration in the high-efficiency zone exceeded 20%.

With a shortage of oil and natural gas, China relies heavily on coal for energy. The coal demand would be more than 50% of the total energy consumption in the long term. Moreover, the energy demand is increasingly aggravated as China is in the stage of rapid development. The demand for coal is bound to increase further if the vigorous development of clean energy is not accelerated. At the same time, the safety problems in the process of coal production have not been effectively curbed. These problems have become an important obstacle to the healthy development of ecological civilization in China [29]. With the westward shift of China's coal mining center of gravity, the proportion of mining of thick coal seams is gradually increasing. The Xinjiang region is rich in coal resources, featuring large coal seam thickness, large coal seam inclination, and high gas content. Due to the limitation of mining technology, a large amount of coal remains in the mining area, easily leading to the abnormal accumulation of gas in the mining area. As an essential means to adjust the movement of the hydraulic bracket in the large inclination coal seam, the PSLs performed a decisive role in the wind flow diameter and flow field distribution at the working area. Meanwhile, as an essential means to solve the gas anomaly in the upper corner, buried pipe extraction has the advantages of a small engineering volume, short working period, simple technology, and sound effects. When the amount of gas gushing from the mining area was small, the buried pipe extraction method could be used alone. When the amount of gas pouring out from the mining area is large, it can be used in combination with other extraction methods. Liuhuanggou Coal mine in Changji area of Xinjiang Province is selected as the research background. In this study, Fluent numerical simulation software was used to study the influence of the length of the slope and the depth of the buried pipe on the gas concentration in mining area and working area. The optimal pseudo-slope length and buried pipe depth were determined and verified in the field. The research results provide important theoretical guidance for gas disaster prevention and control in the upper corner of inclined thick coal seam comprehensive release mining.

## 2. Experimental Design

### 2.1. Overview of the Experimental Working Area

The experimental area used in this study was located in Liuhuanggou Coal Mine, Changji City, Xinjiang Uygur Autonomous Region. Coal was mainly extracted from 4-5 seams. The test working area was mined by the method of long-walled backward caving. The maximum and minimum depth of the orbiting groove on the working area was 557 m and 435 m, respectively. The top and minimum depth of the belt chute was 662.5 m and 486.1 m, respectively. The strike length of the test area is 3250 m, the width of the working area is 180 m, the average coal seam inclination is 24°, and the average mining height is 6.15 m. The mining of inclined coal seams was often accompanied by the problem of upward movement and downward movement of the conveyor. When the slippage was severe, it caused the bracket to dump and could not effectively control the roof, which deteriorated the working area conditions and even caused roofing accidents. At the same time, it also caused the safety exit width of the lower chute to be insufficient, which brought safety risks. The upward movement of the conveyor will cause overload with the conveyor and, in severe cases, will cause the upper chute to be pedestrian. Generally, the PSL is increased to ensure the normal movement of the hydraulic support and to provide more functional space for the headstock. However, the increase in the PSL will affect the gas change between the frames and the upper corner, resulting in abnormal gas concentration at the working area. According to the actual conditions of the test working area, the PSL was adjusted between 18 m and 37 m, respectively. Four groups of 20 m, 25 m, 30 m, and

35 m PSLs were selected for the study. The PSL layout of different working areas are shown in Figure 1.

*2.2. Model Building*

2.2.1. Geometric Modeling

The dimensions of the working area at the site served as the basis for the model for the numerical simulation. The working area inclination length is 180 m. The coal seam inclination angle is 24°. The relic coal seam height is 2 m. The thickness of the direct top was used to compute the theoretical height of the fall zone. Calculate the height of the fall band according to Equation (1).

$$m_z = \frac{h - \Delta}{N_p - 1} \tag{1}$$

where $m_z$ is the height of the bubble fall zone, m; $h$ is the thickness of the coal seam, m; $\Delta$ is the filling thickness due to coal relics, $\Delta = h \times (1 - v) \times N_m$, m; $c$ is the total extraction rate, taken as 85%; $N_m$ is the coefficient of coal breakage and swelling of the fallen top coal, taken as 1.1; and $N_p$ is the coefficient of fragmentation and expansion of coal rock seam, taken as 1.2.

Combining with Equation (1), according to the lithological characteristics of the leading mining area, the theoretical bubble fall zone height of this working area was calculated at 22.8 m. According to Table 1, the range of the fissure zone was 92–130 m, and the height of the fissure zone was taken as 110 m.

**Table 1.** Table of equations for the maximum height of the rift zone.

| Lithology | Applicable to "3 m<$\sum M \leq$12 m" Coal Seam |
|---|---|
| Hard (40~80 MPa, quartz sandstone, limestone, conglomerate) | $H_L = \frac{100 \sum M}{0.15 \sum M + 3.12} \pm 11.18$ |
| Medium-hard (20~40 MPa, sandstone, muddy tuff, shale) | $H_L = \frac{100 \sum M}{0.23 \sum M + 6.10} \pm 10.42$ |
| Soft (10~20 MPa, mudstone, muddy shale) | $H_L = \frac{100 \sum M}{0.31 \sum M + 8.81} \pm 8.21$ |

Note: $M$ is the mining height, m; $H_L$ is the height of the rift zone, m.

According to the lithological characteristics of the main mining area, the height of the theoretical bubble fall zone in the mining area was calculated at 22.8 m and the height of the fissure zone was 110 m, using the above equation.

The workbench-Design Modeler was used to build the geometric model. The mesh module was used to unstructured the model, and the tetrahedral mesh was used to delineate the fluid domain. The geometric model is shown in Figure 2. The model was divided into three different densities of meshes for mesh validation. The total number of grid 1 was 81,371, and the number of working area grids was 7162. The total number of grid 2 was 1,947,318, and the number of working area grids was 400,709. The total number of grid 3 grids was 14,148,963, and the number of working area grids was 3,111,898.

Figure 3 shows the wind speed distribution of nine measurement points arranged along the working area tendency under different grid densities. As the grid density increased, the difference between the results calculated for the density of grid 2 and that of grid 3 was small. Grid extraneous verification showed that the medium-density grid satisfies the irrelevance requirement of the calculation results. The PSLs of 20 m, 30 m, and 35 m were computed using the same mesh-partitioning method.

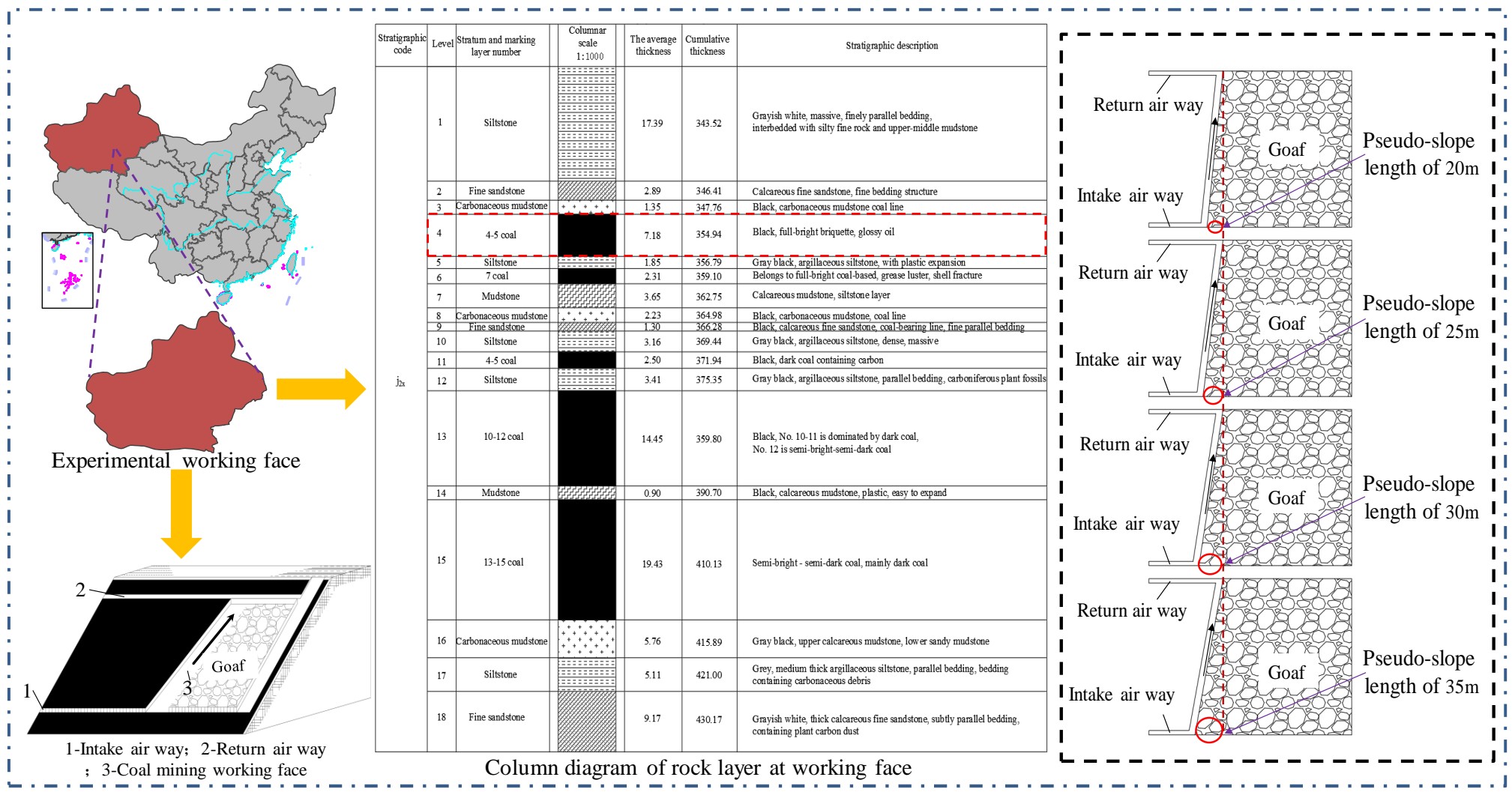

**Figure 1.** Working area layout diagram.

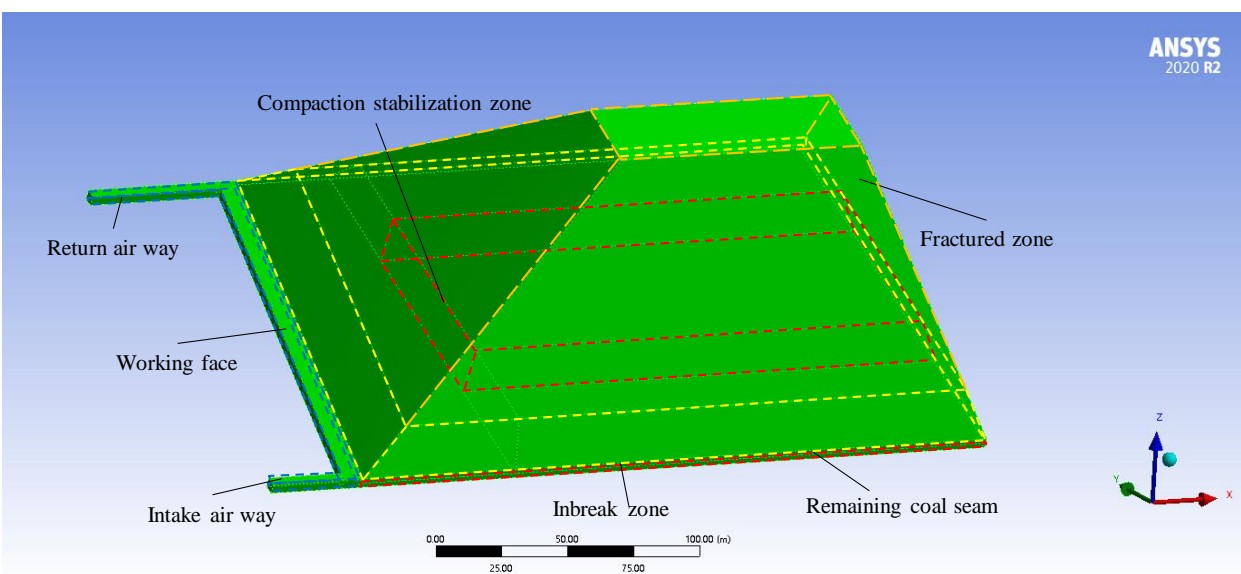

**Figure 2.** Geometric model diagram.

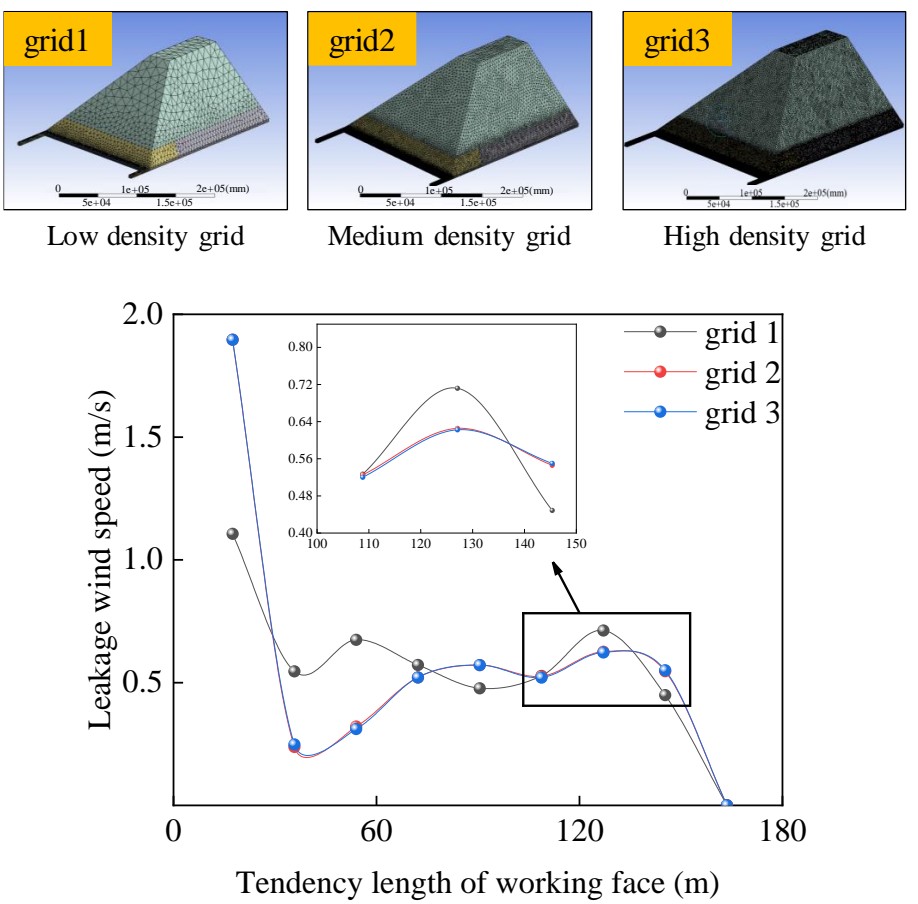

**Figure 3.** Wind speed along the inclination direction of the working area at different grid densities.

2.2.2. Boundary Conditions and Parameter Setting

Several constraints and assumptions must be made to simplify the numerical simulation analysis while ensuring the reliability of the simulation results in the real world. Only methane, nitrogen, and oxygen were considered in the species transport model. In

the computational domain, gas was a continuous and incompressible medium. The flow process did not consider the energy exchanged, such as heat transfer.

(1)    Flow model selection

The Reynolds number was obtained by substituting the field monitoring parameters into Equation (2). The model *RNG-k-ε* was chosen because it added a term to the *ε* equation to improve the accuracy of the flow. Additionally, it provided an analytical formulation of the turbulent Prandtl number and allowed for higher accuracy and confidence compared to the standard *k-ε* model. The fluid was set as a mixed phase of air and methane.

$$Re = \frac{\rho v d}{\mu} \tag{2}$$

where *Re* is the Reynolds number; $v$ is the flow velocity of the fluid, m/s; $\rho$ is the density of the fluid, kg/m$^3$; $\mu$ is the coefficient of viscosity, kg/(m·s); and $d$ is the characteristic length, m.

(2)    Source item settings

After the advancement of the header workings, the relic coal is the primary source of gas gushing out from the mining area. Under the influence of mining, the gas gushed out from the adjacent coal seam under the main mining seam after unloading pressure. Finally, it diffused and floated through the legacy coal seam to the mining area. Integrate the gas outflow from the coal deposit, the adjacent seam, and the surrounding rock to the source of porous media in the bottom 2 m of the model. It was calculated that the total gas outflow from the mining area was 6.15 m$^3$/min, and the gas quality source term in the mining area was calculated according to Equation (3).

$$Q_v = \frac{Q_g \rho_g}{V} \tag{3}$$

where $Q_v$ is the gas mass source term gushing volume, kg/(m$^3$·s); $Q_g$ is the absolute gas gushing volume, m$^3$/s; $\rho_g$ is the gas density, 0.716 kg/m$^3$; $V$ is the model volume of the mining area, m$^3$; and $Q_v = 6.15 \times 0.716/(2.7 \times 10 - 6) = 1.6e^{-6}$kg/(m$^3$·s).

(3)    Basic flow equations [30,31]

① Conservation of mass Equation:
Equation (4) depicts the mass conservation equation, also known as the continuity equation.

$$\frac{\partial \rho}{\partial t} + \nabla(\rho v) = S_m \tag{4}$$

where $t$ is the time, s; and $S_m$ is the mass added to the continuous phase by the dispersive secondary term and the defined source, kg/(m$^3$·s).

② Conservation of momentum equation:

$$\frac{\partial}{\partial t}(\rho v) + \nabla(\rho v v) = -\nabla p + \nabla(\tau) + \rho g + F \tag{5}$$

where $p$ is the static pressure, Pa; $\tau$ is the stress tensor, Pa; $\rho g$ is the gravitational force; and $F$ is the external force, N.

③ Component mass conservation equation:

$$\frac{\partial}{\partial t}(\rho c_s) + \nabla(\rho v c_s) = \nabla(D_s \mathrm{grad}(\rho c_s)) + S_s \tag{6}$$

where $c_s$ is the volume fraction of component s; $D_s$ is the diffusion coefficient, m$^2$/s; and $S_s$ is the mass of the component produced by the chemical reaction per unit time, kg/(m$^3$·s).

④ Considered the extraction area as a porous medium and add momentum sources, which included viscous loss term and inertial loss term.

$$S_i = \sum_{j=1}^{3} E_{ij}\mu v_j + \sum_{j=1}^{3} F_{ji}\frac{1}{2}\rho v_{\mathrm{mag}}v_j \tag{7}$$

where $S_i$ is the source of the "$i$" ($x$, $y$, $z$) momentum equation, N/m$^3$; $\mu$ is the molecular viscosity; $v_{\mathrm{mag}}$ is the mode of the velocity vector; $v_j$ is the velocity component in the "$x$, $y$, $z$" direction, m/s; and $E$ and $F$ are predefined matrices.

⑤ The type of flow in the model is turbulent, and the solution equations are given in Equations (8) and (9) using the "*RNG-k-ε*" model in Fluent.

$$\frac{\partial}{\partial t}(\rho k) + \frac{\partial}{\partial x_i}(\rho k u_i) = \frac{\partial}{\partial x_i}\left(\alpha u_{iff}\frac{\partial k}{\partial x_j}\right) + G_{\mathrm{k}} + G_{\mathrm{b}} - \rho\varepsilon - Y_{\mathrm{M}} + S_{\mathrm{k}} \tag{8}$$

$$\frac{\partial}{\partial t}(\rho\varepsilon) + \frac{\partial}{\partial x_i}(\rho\varepsilon u_i) = \frac{\partial}{\partial x_j}\left(\alpha_\varepsilon u_{eff}\frac{\partial\varepsilon}{\partial x_j}\right) + G_{1\varepsilon}\frac{\varepsilon}{k}(G_{\mathrm{k}} + C_{3\varepsilon}G_b) - C_{2\varepsilon}\rho\frac{\varepsilon^2}{k} - R_\varepsilon + S_\varepsilon \tag{9}$$

where $G_{\mathrm{k}}$ is the turbulent kinetic energy generated by the velocity gradient, J; $G_{\mathrm{b}}$ is the turbulent kinetic energy generated by buoyancy, J; and $k$ is the turbulent kinetic energy, J. $Y_{\mathrm{M}}$ is the contribution of pulsating expansion in compressible turbulence to the total dissipation rate; $\alpha_{\mathrm{k}}$, $\alpha_\varepsilon$ are the inverse effective Prandtl numbers of "$k$", "$\varepsilon$", respectively; $S_{\mathrm{k}}$, $S_\varepsilon$ are the defined source terms; $k$ source term in kg/m·s$^3$; $\varepsilon$ source term in kg/m·s$^4$; and $C_{1\varepsilon}$, $C_{2\varepsilon}$, and $C_{3\varepsilon}$ are constants.

(4)　Porosity setting of mining area module

The mining area model was divided into two modules, fracture zone, and fallout zone. The porosity of the fracture zone was divided according to the "O" circle theory, and the porosity was calculated according to Equation (10) [32]. The average porosity $P_0$ distribution of the sub-domains was shown in Table 2.

$$\varphi(x,y) = 1 + \frac{\left(1 + e^{-0.15*\left(\frac{l_y}{2} - |y|\right)}\right)\left[1 - \frac{h_{\mathrm{d}}}{h_{\mathrm{d}} + H - \left(H - h_{\mathrm{d}}\left(N_{\mathrm{P_b}} - 1\right)\right)\left(1 - e^{-\frac{x}{2l}}\right)}\right] - 1}{1 + \sigma_0^{-1}\beta_1\gamma\left(\frac{l_y}{2} - y\right)\sin\theta} \tag{10}$$

where $l_{\mathrm{y}}$ is the tendency width of the mining area, m; $h_{\mathrm{d}}$ is the thickness of the direct top, m, take 3 m; $N_{\mathrm{P_b}}$ is the coefficient of rock fragmentation and expansion after the direct top damage, take 1.1; l is the length of the basic fixed broken rock, m, take 15 m; $\gamma$ is the bulk weight of the fallen rock, N/m$^3$, the bulk weight of the fallen gangue is generally $2 \times 10^4$~$3 \times 10^4$; $\beta_1$ is the regression coefficient, the fallen rock is Mudstone is taken $-0.028$; and is the coal seam inclination, 24°.

**Table 2.** Porosity assignment table for each part of goaf.

| | Remaining Coal Seam | Inbreak Zone | | | | Fractured Zone |
| --- | --- | --- | --- | --- | --- | --- |
| | | Natural Accumulation Area | Influence Area of Coal Wall on the Return Wind Side | Compaction Stabilization Zone | Inlet Side Coal Wall Influence Area | |
| $P_0$(%) | 0.2 | 0.4 | 0.25 | 0.13 | 0.22 | 0.2 |

(5)　Parameters of the inlet and return air lanes

The airflow was inflowed from the inlet lane. The inlet air speed was 3.5 m/s. The inlet air mass fraction was 23%, which was converted to a molar fraction for calculation

purposes. The return air lane was set as a free outlet. The hydraulic diameter was taken as the equivalent diameter of the lane. The calculation was made according to Equation (11).

$$de = \frac{2ab}{(a+b)} \tag{11}$$

where *de* is the equivalent diameter of the roadway, m; and *a* and *b* are the geometric length and width, m, respectively.

### 2.3. Monitoring Point Layout Program

Gas concentration monitoring at the working area was calculated via grid nodes. Gas concentration was monitored by arranging to measure surfaces, measuring lines, and measuring points in the calculation domain. The working area domain was divided into three measurement surfaces that were parallel to the base plate, and the measurement points on each surface were organized using the point cloud measurement method. The number of point clouds was set to 1000. There was a total of 12 measuring lines that were spaced 0.3 m, 2.8 m, 5.3 m, and 7.8 m from the coal wall, respectively. Each group of measurement lines was separated into upper, medium, and lower spatial heights. At the end of the return air tunnel, we set up a 2 m by 2 m by 2 m grid and nine grid measurement stations to keep an eye on the gas concentration in the upper corner. The measurement arrangement was shown in Figure 4.

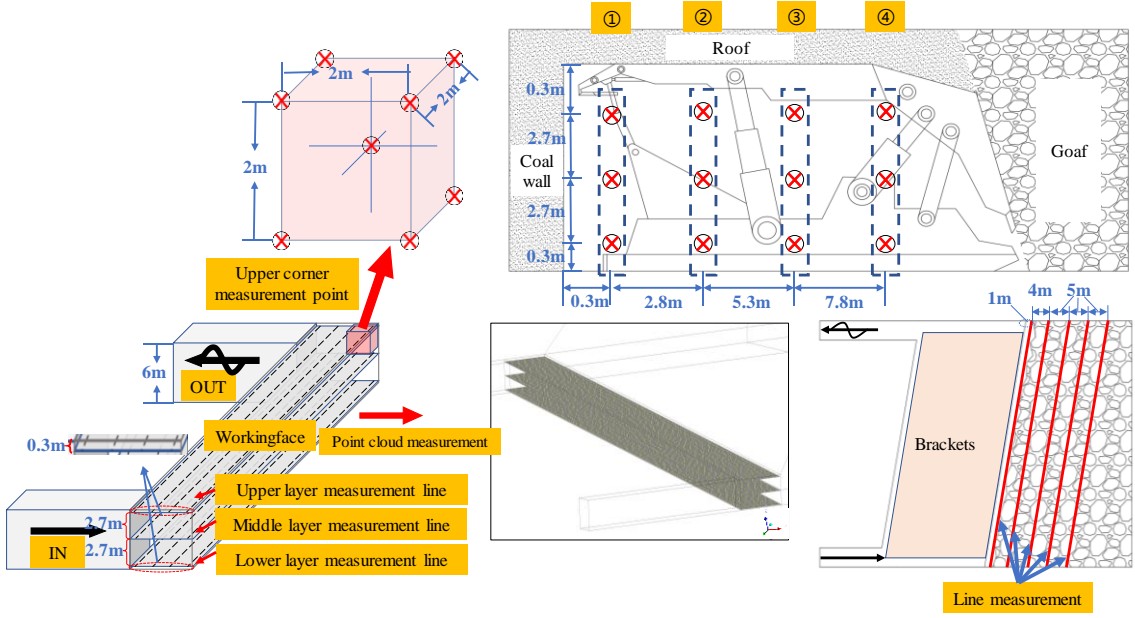

**Figure 4.** Working area measuring point layout.

## 3. Gas Concentration Distribution Pattern under Different PSLS

### 3.1. Variation in the Working Area's Gas Spewing Location while Using Various PSLs

The flow field vector lines at each location inside the mining region are shown in Figure 4 under the influence of various PSLs, respectively. Under various PSLs, the flow field lines in the mining region exhibited significant variances. The wind in the mining region flowed into the working area from the upper corner when the PSL was 20 m and 25 m. When the PSL was 30 m, two air leakage spots on the working area were visible, one in the A region and the other in the upper corner. The B area and the top corner of the working area were the two primary air leakage locations when the PSL was 35 m, with the B area being closer to the inlet tunnel than the A area, as shown in Figure 5a–d. In summary, as the PSLs increased to more than 30 m, the air leakage point drifted toward the

inlet tunnel. The wind transported high gas concentrations from the mining region to the working area, abnormally concentrated at the wind leakage area.

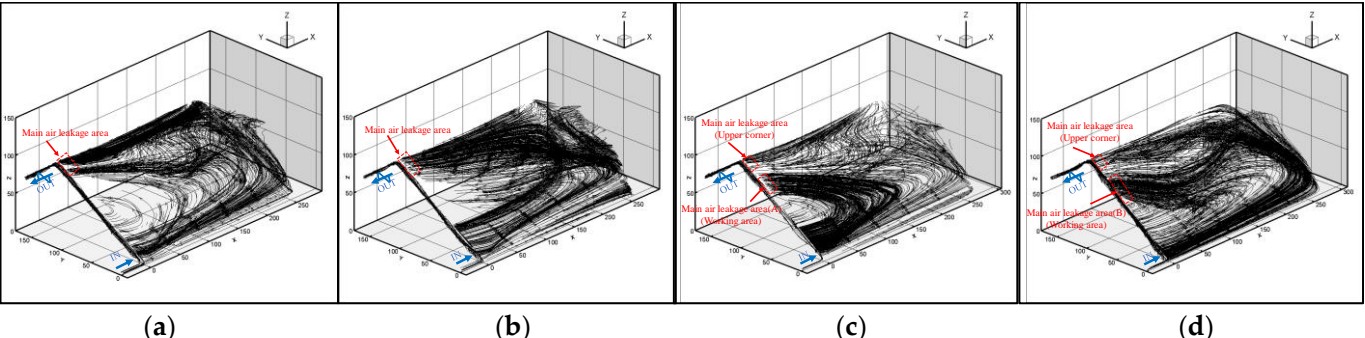

|  (a)  |  (b)  |  (c)  |  (d)  |

**Figure 5.** Airflow vector map of goaf under different PSLs. (**a**) PSL-20 m; (**b**) PSL-25 m; (**c**) PSL-30 m; (**d**) PSL-35 m.

The disorder and uniformity in the arrangement of measurement points characterize the three-dimensional grid point cloud measurement method. The experimental results show that the measured gas concentration can reflect the spatial distribution of gas concentration under different PSLs.

Figure 6 shows the analysis of gas concentration distribution in working areas at different levels. The experimental results showed that PSLs have a great influence on the location of gas accumulation zone in the working area (concentration of gas accumulation zone ranges from 0.8% to 1.0%, and over 1.0% of the abnormal gas zone). The overall gas concentration in the working area was low, and gas concentrated in the upper corner under PSL-20 m, PSL-25 m, and PSL-30 m. At a PSL of 35 m, gas accumulated in the upper corner and at the working area. Gas builds up in the PSL's 20 m and 25 m upper corners at the middle and upper levels. Gas builds up at the upper corner and central portion of the working area at PSL of 30 m and 35 m. The working area's gas concentration gathering region was moved nearer the input tunnel as the PSLs increased. The gas accumulation zones (expressed as the distance from the inlet lane, m) and peak gas concentration (%) at different PSLs' working areas are shown in Table 3.

**Table 3.** Distribution of gas anomalous areas and gas concentration peaks in working area under different PSLs.

|  | PSL 20 m | | PSL 25 m | | PSL 30 m | | PSL 35 m | |
|---|---|---|---|---|---|---|---|---|
|  | Gas Concentration Anomaly Zone (m) | Maximum Gas Concentration (%) | Gas Concentration Accumulation Area (m) | Maximum Gas Concentration (%) | Gas Concentration Anomaly Zone (m) | Maximum gas Concentration (%) | Gas Concentration Anomaly Zone (m) | Maximum Gas Concentration (%) |
| Upper Level | 84.9~159.7 | 1.76% | 156.4~160.6 | 0.98% | 64.1~161.1 | 3.05% | 23.6~163.1 | 4.90% |
| Middle Level | - | - | 160.1~161.1 | 0.81% | 65.3~73.6 149.4~160.1 | 1.46% 1.58% | 29.9~126.6 155.4~163.1 | 3.28% 2.57% |
| Lower level | - | - | - | - | - | - | 35.2~102.3 160.2~161.7 | 2.19% 1.88% |

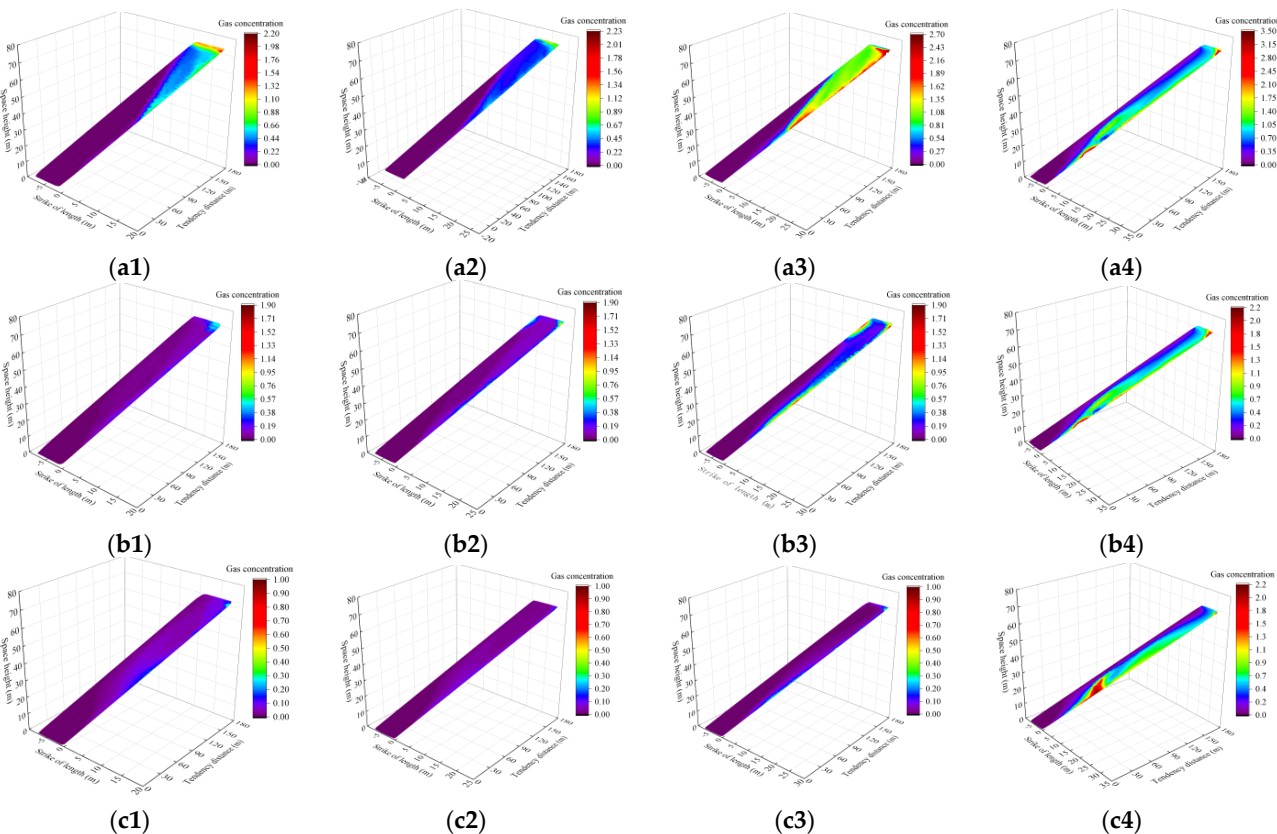

**Figure 6.** Gas cloud map of the working area under different PSLs. (**a1**–**a4**) show the gas concentration clouds at the high level of the working area at pseudo-slope lengths of 20 m, 25 m, 30 m, and 35 m, respectively; (**b1**–**b4**) show the gas concentration clouds at the middle level of the working area at pseudo-slope lengths of 20 m, 25 m, 30 m, and 35 m, respectively; (**c1**–**c4**) show the gas concentration clouds at the low level of the working area at pseudo-slope lengths of 20 m, 25 m, 30 m, and 35 m, respectively.

In order to reflect the spatial distribution of gas concentration in the working area, four groups of measuring lines are arranged according to the working area. According to Formula (12), the standard deviation of gas concentration at different positions in the tilt direction of the working area was calculated. The gas concentration error zone diagram in the working area was drawn, as shown in Figure 7a–d. The gas concentration error band chart reflected the variation characteristics of the spatial trend average gas concentration data supported by the comprehensive working area. According to the change rate of average gas concentration and gas anomaly in the working area, the gas concentration curve in the working area was divided into three zones. $G_0$ was the gas concentration area of the working area, $G_1$ was the gas growth area of the working area, $G_2$ was the gas anomaly area of the working area. Zone $G_0$ was located near one side of the intake tunnel. As the dip Angle of coal seam was large and the space height of air inlet roadway was low, the gas in the mining area will be transported to the side of air return roadway under the action of floating. A small amount of gas overflowing from the mining area to the working area will be quickly carried away by fresh air with high velocity, so the gas area at the side of the intake lane was 0. In $G_1$ zone, the spatial height of the working area increased, resulting in higher gas concentration in the mining area. Due to the influence of ventilation resistance, the wind speed in the working area decreased and the dilution effect on gas was weakened, resulting in the increase in gas concentration in the area. In the space 7.8 m away from the coal wall, there exists the $G_2$ zone, which was an abnormal gas concentration zone in the working area, and most of the gas gush from the mining area will be diluted by fresh air and taken out of the working area. However, when a large amount of gas gushing out,

the flow of wind cannot dilute the gas in time, leading to the occurrence of gas anomalies in different positions of the working area far from the coal wall.

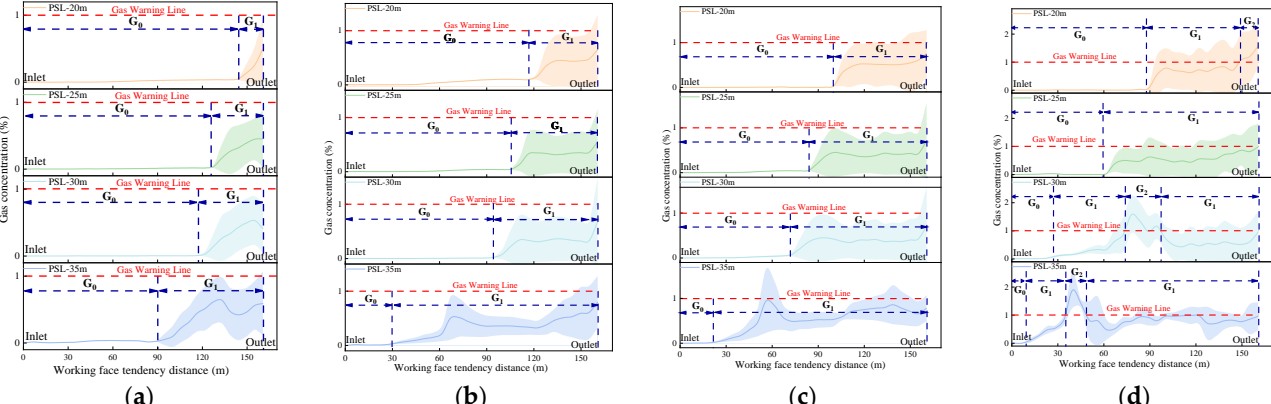

**Figure 7.** Distribution of Gas Concentration in Space with Different PSLs. (**a**) Distance to coal wall 0.3 m. (**b**) Distance to coal wall 2.8 m. (**c**) Distance to coal wall 5.3 m. (**d**) Distance to coal wall 7.8 m.

Under different PSLs conditions, the working area's gas distribution pattern differed. Overall, the $G_0$ zone extended further away from the coal wall. The range of the $G_0$ zone gradually decreased as the PSLs increased, while the scope of the $G_1$ zone gradually expanded toward the side of the inlet tunnel. Different PSLs factored occasionally had gas concentration abnormalities at 0.3 m and 2.8 m from the coal wall, but the average gas concentration value did not exceed 1%. A gas concentration of 0.94% at 5.3 m from the coal wall, PSL of 35 m, and 56 m, respectively, from the inlet airway. The $G_2$ zone appeared in the upper corner at 7.8 m from the coal wall and a PSL 20 m. When the PSL was 30 m or 35 m, the $G_2$ zone appeared on the working area, which was 78.1 m and 45.1 m away from the incoming wind tunnel, respectively. The gas concentration at different heights in inclined coal seams varied greatly due to the wind flow's dual action and the gas's rise and float. When the PSL was 20 m, 25 m, or 30 m, the vertical distribution of gas was more stable at 0.3 m, 2.8 m, and 5.3 m from the coal wall, respectively. When the PSL was 30 m, it was 7.8 m from the coal wall, and the vertical distribution of gas in the $G_1$ area was less stable in the range of 61.6~91.6 m from the inlet airway. When the PSL was 35 m, the vertical distribution of gas concentration in the $G_1$ area was disordered. An abnormal area of gas concentration appeared due to severe air leakage from the working area. The tendency lengths of gas anomalies that appeared in the working area under different PSLs were fitted to obtain the spatial distribution of gas in the working area under different PSLs, as shown in Figure 8. The average gas area of the working area was below the curve, and the mining area's gas-gushing influence area was above the curve. When the PSL was 20 m and 25 m, the gas gushing out of the mining area impacted the area dividing the line in line with the linear distribution, as shown in Equations (12) and (13). When the PSL was 30 m and 35 m, the high-concentration gas gushed from the mining area to the working area in advance. At this point, the gas gushing out of the mining area was influenced by the area dividing line in line with $y = y_0 + Ae^{-x/t}$, as shown in Equations (14) and (15).

$$Y = 144.66478 - 7.2749X \tag{12}$$

$$Y = 130.77093 - 8.5809X \tag{13}$$

$$Y = 119.73454exp(-X/5.29068) + 2.82769 \tag{14}$$

$$Y = 112.10118exp(-X/1.35962) + 15.60497 \tag{15}$$

where *Y* is the distance from the gas emergence point to the inlet tunnel, m; and *X* indicates the distance from the coal wall, *m*.

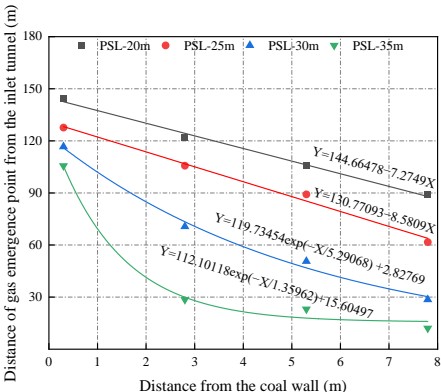

**Figure 8.** Spatial distribution map of the gas-containing area in working area under different PSLs.

When the PSLs were short, the mining area's air leakage point was on the return wind side. The wind flow carries away a small amount of gas gushing out from the working area, and the gas concentration at the working area was 0. The area of the gas emergence domain at the working area was calculated by integration as $W_{20m}$ = 575.99 m², $W_{25m}$ = 719.86 m², $W_{30m}$ = 965.27 m², and $W_{35m}$ = 1201.22 m². The results showed that as the PSLs increased, as did the area affect by gas gushing from the working area.

## 3.2. Gas Concentration Distribution in the Upper Corner at Various PSLs

Most of the fresh air used in coal mining operations entered the working area from the inlet roadway and carried the coal wall and support overflow gas out from the return air side. The high concentration of gas was brought into the work by the small air flow from the return air side, and the small air flow entered the mining area from the intake side. In the working area, the fluid domain was turbulent, so it was easy to generate eddy currents at the angle between the working area and the road surface. In the upper corner, the vortex created an airflow blind spot, encouraging the gas to build up, causing gas anomalies. Under different PSLs, the upper corner gas concentration fluctuated more obviously, as shown in Figure 9. Figure 9 shows the error bands for gas concentrations determined by nine measuring points located on different PSLs in the upper corners of the layout. The solid line at the top corner shows the average gas concentration. The dashed line was the fitting curve of upper corner gas concentration increasing with PSLs. Error tape was the shaded area. With the increase in PSLs, the upper corner gas concentration decreased gradually. The average gradient of gas concentration in the upper corner of PSLs was 0.0856%/m, 0.010%/m, and 0.0066%/m, respectively, in the range of 20~35 m. The inflection point of airflow of different PSLs was different in mining area. The results showed that the leakage area and the upper corner vortex intensity vary with the pressure difference. In other words, with the increase in PSLs, the size of the shadow region gradually increased, and the concentration of each measuring point fluctuated obviously. As mentioned above, the upper corner gas concentration gradually decreased as the PSLs increased. However, at 35 m PSL, the average gas concentration in the upper corner still reached 0.89%, and the gradient of the decrease in the upper corner gas concentration decreases gradually. Therefore, there is an effective range for PSLs to reduce the upper corner concentration. The increase in PSLs would increase the length of the working area, reduce the area of the extraction area, increase the gas concentration in the extraction area, and leads to gas gushing out of the working area. The effect of reducing the upper corner gas concentration became progressively worse. Therefore, it was of great significance to find the optimal PSLs for gas disaster management in the working area.

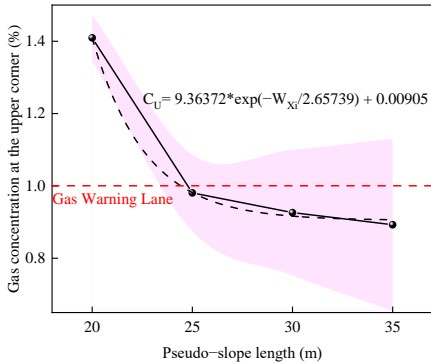

**Figure 9.** Gas concentration in the upper corner under different PSLs.

## 4. The Effect of the Coupling Mechanism of the BPDs of the Extraction Port and PSLs on the Gas Concentration Distribution in the Working Area

*4.1. Influence of BPDs and PSLs on the Gas Concentration Distribution in the Extraction Zone*

The gas concentration on the return side of the "U" ventilation of the inclined coal seam was high. The working area might have had abnormal gas concentration. Although increasing the air volume at the working area can reduce the gas concentration to a certain extent. Due to site conditions, equipment and operation costs, and wind speed limitations, the air volume cannot be increased indefinitely. Therefore, if wind discharged alone cannot solve the problem of gas overload, effective extraction measures must be taken. Buried pipe extraction, as an auxiliary gas extraction method, was used to solve the problem of gas overload in the upper corner. The specific practice was to lay a large diameter steel pipe in the outer helper of the return airway near the bottom plate. A tee containing a combination of valves was installed on the pipe at regular intervals. The gas extractor was connected when the working area was advanced to a certain distance. The top of the extractor was inserted vertically into the roof plate. When it was in the best extraction position in the extraction area, the valve of the extraction port was opened to extract gas from the extraction area. When the extraction port continued to penetrate deeper into the extraction zone, the next three-way valve was opened so that the extraction port was always in the best extraction position. The gas in the upper corner was often abnormal for the high gas concentration on the return side of the "U" ventilation. The effect of BPDs and PSLs on the gas concentration distribution in the extraction area was simulated by Fluent. Figure 10 shows the geometric model of the extraction zone for buried pipe extraction. According to the site situation, the buried pipe extraction pipe was proposed to be a 0.4 m cylindrical pipe. The height of the BPD was 2.6 m from the roof, and the negative pressure of the buried pipe extraction port was 8 kPa.

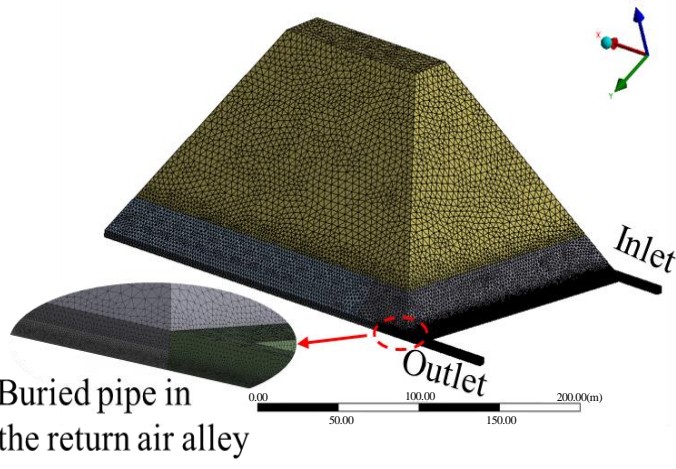

**Figure 10.** Geometric model of the extraction zone under buried pipe extraction.

To study the influence of BPDs and PSLs on gas concentration distribution, Fluent was used to simulate BPDs of 10 m, 20 m, 30 m, and 40 m at PSLs of 20 m, 25 m, 30 m, and 35 m, respectively. The complete results are shown in Figure 11a–d.

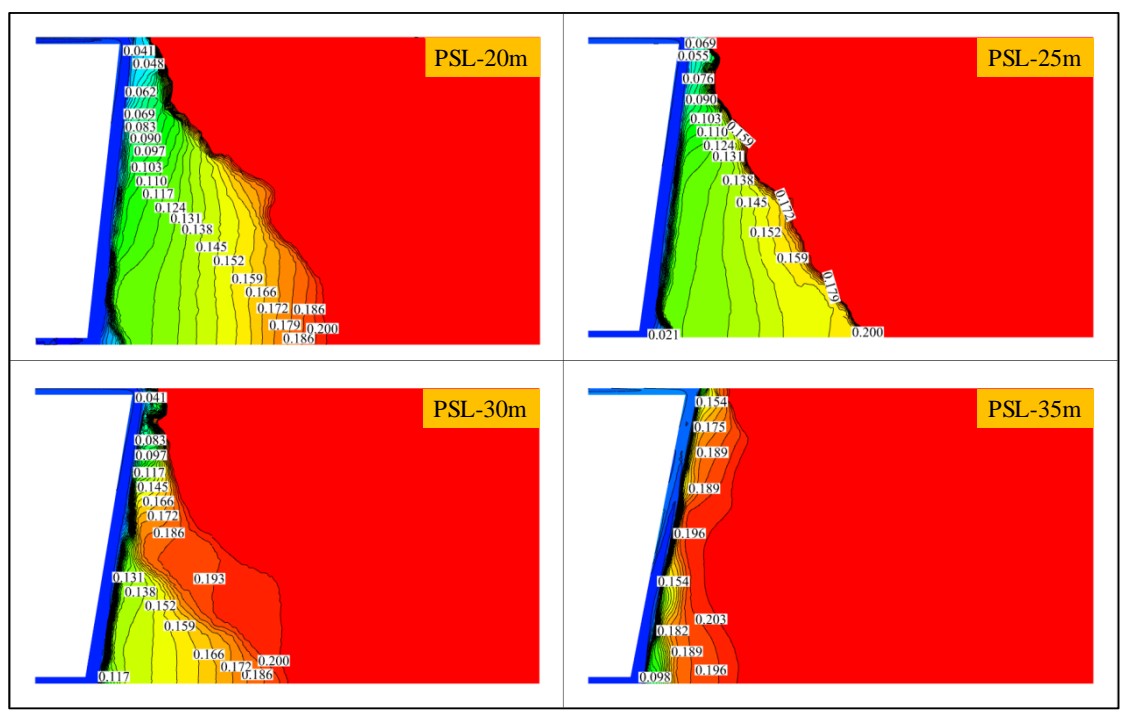

(**a**)

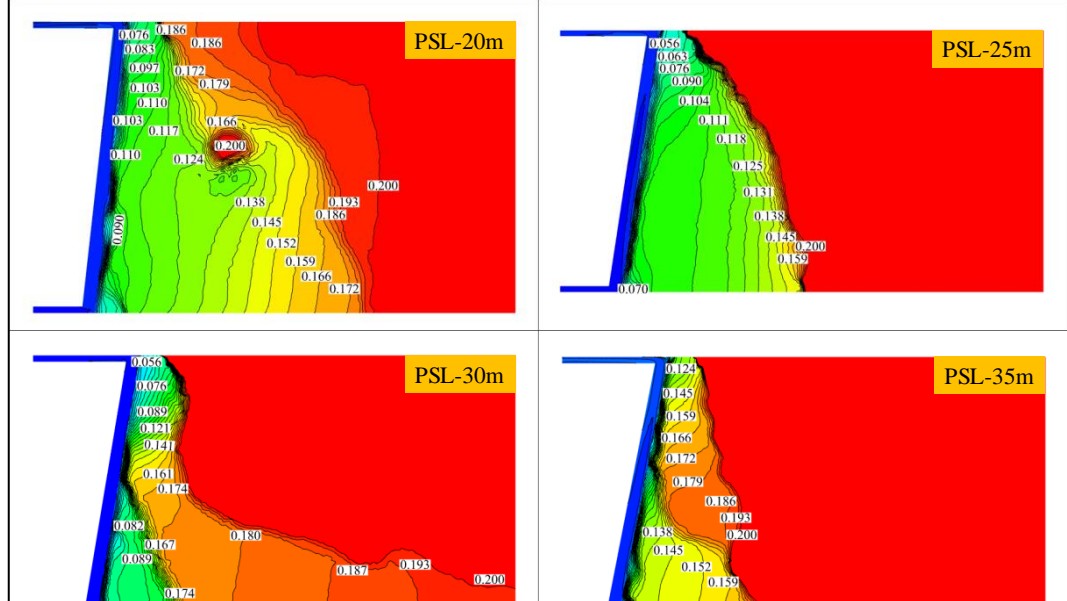

(**b**)

**Figure 11.** *Cont.*

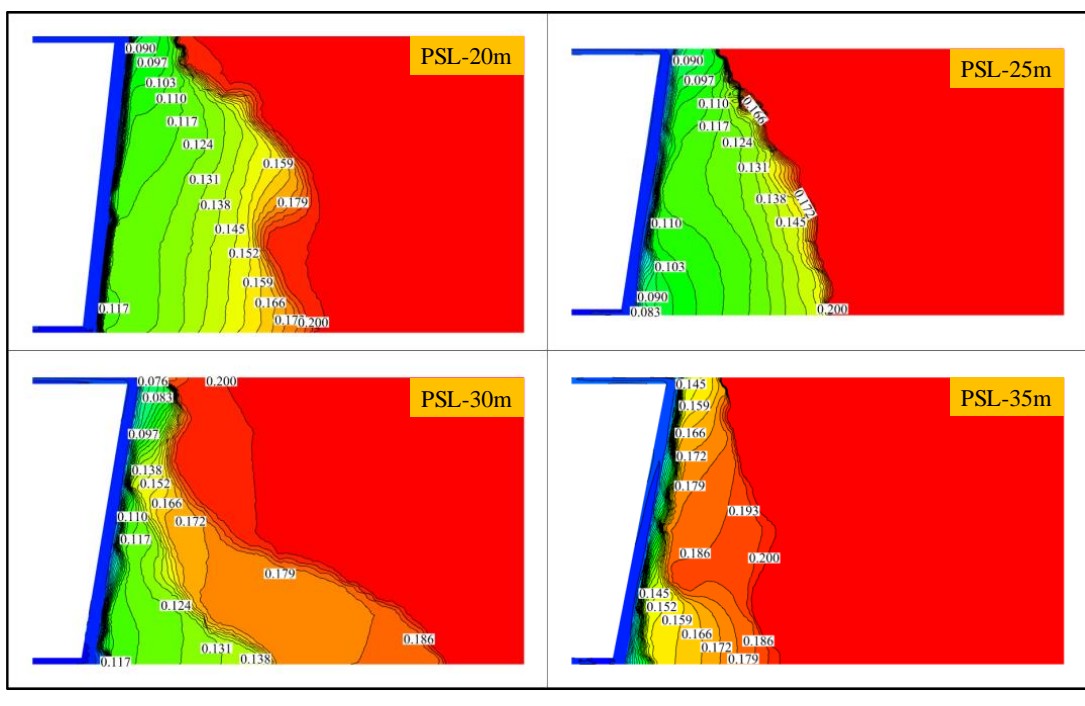

(**c**)

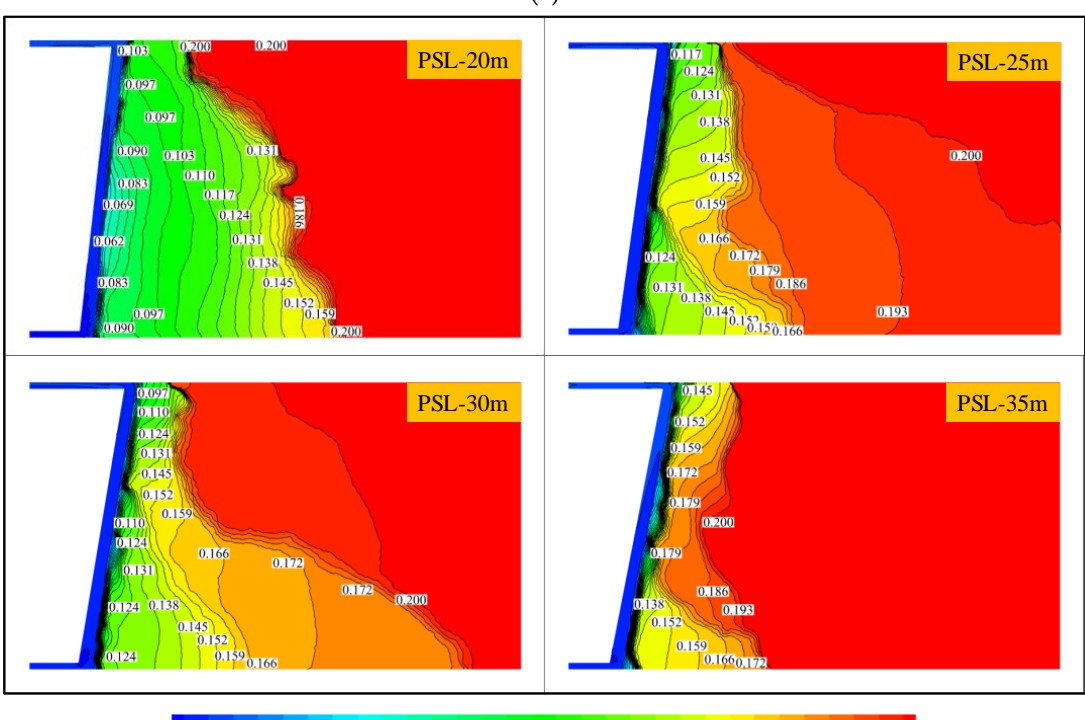

(**d**)

**Figure 11.** Cloud map of gas concentration distribution in the extraction zone under the coupling effect of different PSLs and BPDs: (**a**) BPD—10m; (**b**) BPD—20m; (**c**) BPD—30m; (**d**) BPD—40m.

Figure 12 shows the simulation results of gas concentration distribution at different PSLs and BPDs in the roof height of the return air tunnel. When the PSL was 20 m, the

influential zone of the inlet airway and the suitable location of the buried pipe extraction under different BPD conditions were all through in the mining area, forming a low gas area channel. When the PSL was 25 m, the influential area of the incoming wind lane and the adequate size of the buried pipe extraction failed to form a penetration when the BPD was 40 m, and the high concentration gas area in the mining area spread to the working area. When the PSL was 30 m and 35 m, the influential zone of the inlet airway and the suitable location of the buried pipe extraction failed to form a penetration under different BPDs, and the high gas concentration area in the extraction area tended to spread to the working area. With the increase in BPDs, the low-concentration gas area within the extraction radius in the A (buried pipe influence area) area gradually increased. However, influenced by the PSLs, the location of low gas concentration in the B (wind-influenced place) area gradually decreased. When the PSL was 25 m and the BPD was 40 m, the penetration of the air inlet influence zone and buried pipe extraction influence zone in the mining area was interrupted, and high gas appeared in the middle of the two zones and gathered near the working area. A measuring line was arranged at the height of the top plate in the return tunnel to monitor the gas concentration distribution along the return side direction.

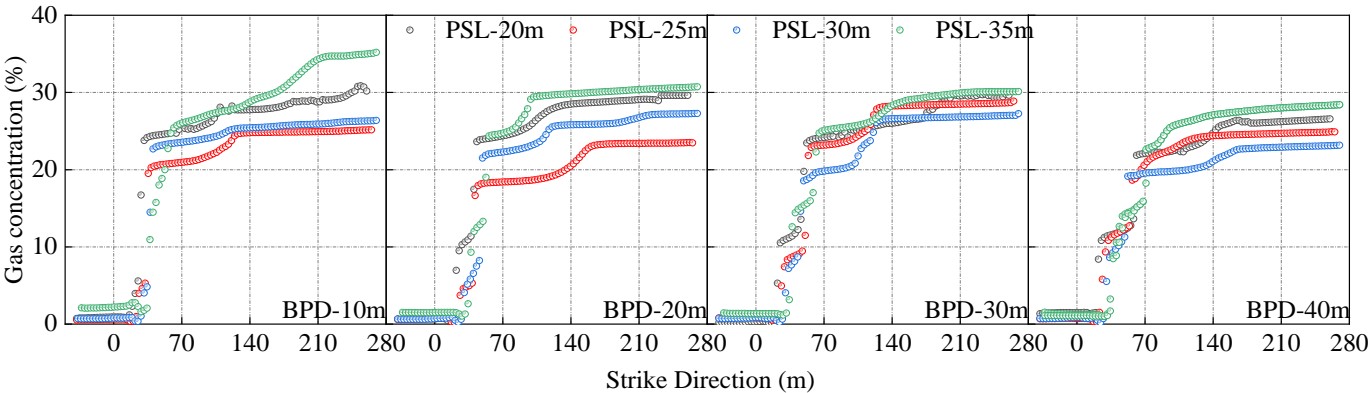

**Figure 12.** Gas concentration distribution pattern on the return side.

When comparing the gas concentration extraction on the inlet and return side of the mining area under the joint action of PSLs and BPDs, the results showed that the extraction port was closer to the working area, affected by air leakage, and could not maximize the negative pressure, and it was easy to form a vortex on the return side. When the buried pipe extraction port was 10 m from the working area, the gas concentration on the return wind side decreased rapidly in the range of 22~58 m and slowly in the field of 58 m~270 m. In the case of the same burial depth of the extraction port, the gas concentration on the return wind side decreased, then increased when the PSL increased. The negative pressure of the buried pipe has limited influence on the deep part of the compacted mining area. It can only extract gas locally, so its concentration remained unchanged within a certain distance. With the increased PSLs and the BPDs, the joint action of the leakage wind flow and damaging pressure extraction wind flow tended to be smooth, and the low-concern gas forms a penetration near the working area. The unloaded gas was transported to the return wind side under the influence of wind flow and discharged by the negative pressure of the BPD. As the extraction port's depth increased, the buried pipe extraction's guiding effect decreased. The low gas penetration area formed by the negative extraction pressure and the leakage wind flow gradually disappears, and the high concentration gas accumulates near the working area. Although the gas concentration in the direction of the return wind side was low, the gas management of the upper corner has yet to play a noticeable effect. Therefore, when BPDs and PSLs work together, the PSL was challenging to exceed 25 m, and the BPD-20 m was more reasonable.

### 4.2. Influence of the Combined Effect of BPDs and PSLs on the Distribution Pattern of Gas Concentration in the Upper Corner

To study the pattern of high-concentration gas gushing out from the mining area to the working area under different PSLs and BPDs. The fourth set of measurement lines monitored the gas concentration distribution at the working area near the extraction zone. The test results were shown in Figure 13a–d, where different colors were blocked to represent different gas concentration intervals.

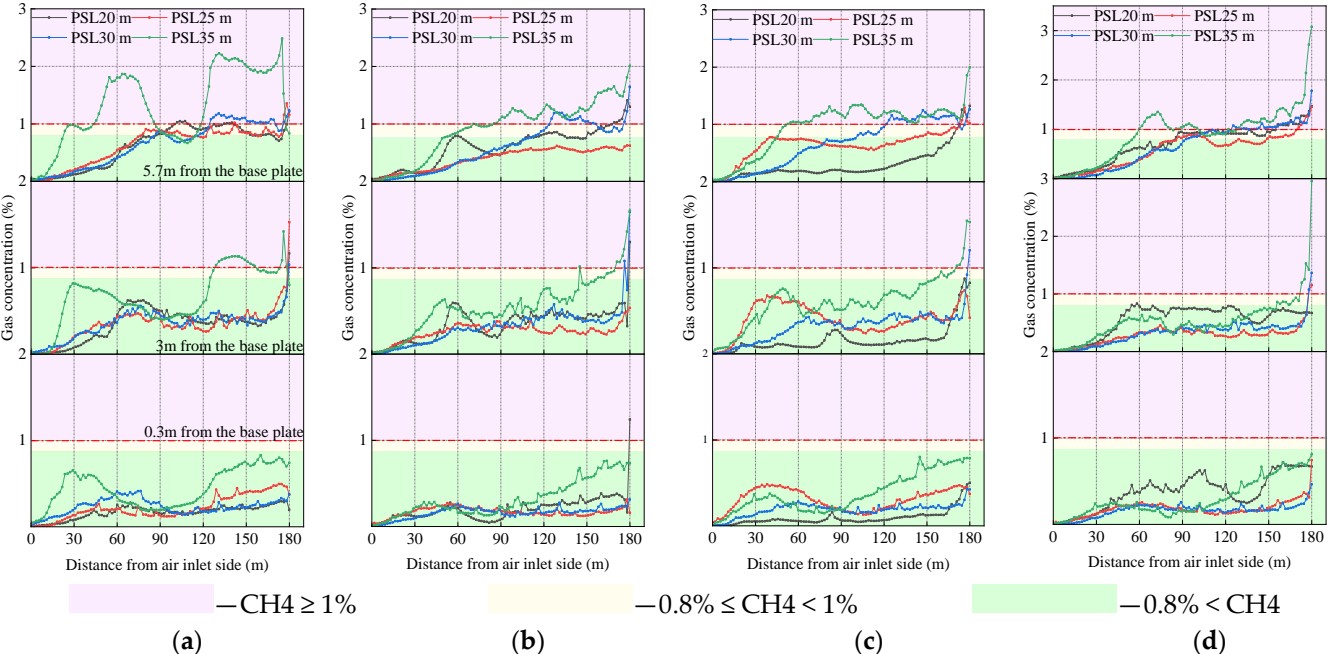

**Figure 13.** Distribution of gas concentration in the working area of the near mining area. (**a**) BPD—10 m; (**b**) BPD—20 m; (**c**) BPD—30 m; (**d**) BPD—40 m.

Gas exhibits significant variability in the spatial height of the working area under the action of rising floats. When the extraction port was 10 m from the working area, the high concentration of gas in the deep part of the extraction area gathered near the working area under the influence of the negative pressure of buried pipe extraction. Under the intervention of air leakage, the gas in the extraction area gushed out to the working area. With the increase in PSLs, the gas concentration at the working area also increased gradually, and there were more gas concentration abnormalities at the working area when the PSL was 35 m. When the extraction port was buried 20 m deep, gas did not accumulate at the working area when the PSL was 20 m. With the increase in the extraction opening depth, the extraction effect of the upper corner gradually decreased. Due to the guiding influence of wind flow, gas accumulation occurs in the upper corner under different working area PSLs and gas accumulation occurs in the working area.

Figure 14 shows the effect of different PSLs and BPDs on the concentration of extracted gas. Due to the low concentration of gas gathering on the return side at the PSLs of 25 m and the influence of air leakage, the air leakage diluted and carried away. Therefore, the extraction concentration was the lowest at the PSLs of 25 m for different extraction opening depths. When the extraction port was buried 10 m deep, the extraction effect was poor, and the concentration of extracted gas was low, about 13%. The extraction port in this area was shallow and influenced by the leakage wind flow, so the gas was diluted and carried away, which led to low extraction concentration and could not effectively prevent the gas in the mining area from gushing out to the working area. When the extraction port was buried 20 m deep, the extraction concentration reached 17%. The effect of managing the abnormal gas accumulation in the corner of the working area was remarkable. The gas

concentration in the upper corner was the lowest, about 0.43%, as shown in Figure 15. When the extraction port was 30 m inside the extraction area, the extraction gas concentration was more significant, about 20%. Although the gas extraction concentration was higher in the extraction area, the effect of extracting gas on the corner angle on the working area was gradually reduced. When the extraction port was just 40 m inside the extraction area, the extraction concentration was as high as 28%. However, there was an abnormal gas concentration in the upper corner because the extraction port was far from the working area. Although the extraction concentration was high, it had less effect on the gas control of the working area. The results were consistent with Liu. Z's [33] on the effect of BPD on gas concentration in the upper corner.

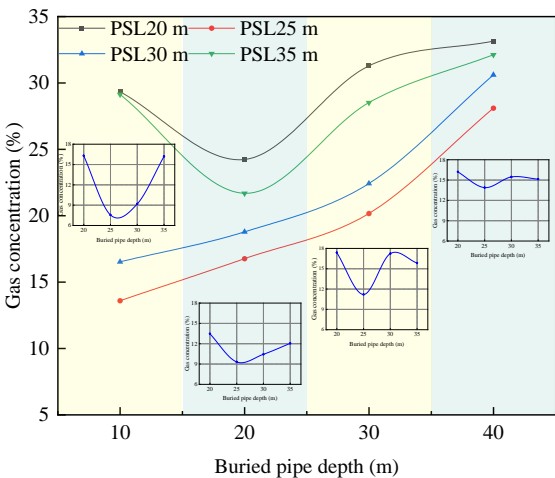

**Figure 14.** Variation pattern of extracted gas concentration.

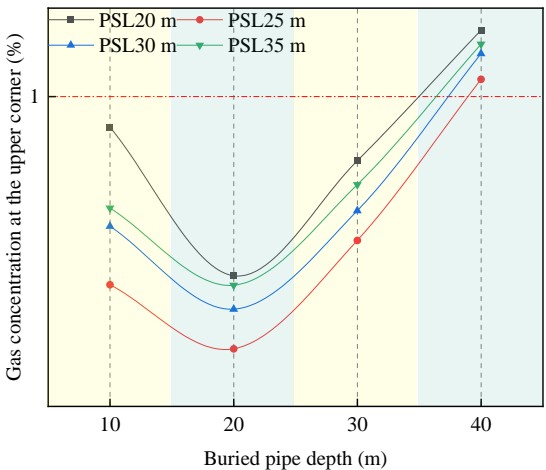

**Figure 15.** Change pattern of gas concentration in the upper corner.

In sum, at PSL 25 m and BPD 20 m, the influencing area on the inlet side was connected with the influencing area of buried pipe extraction. There was no gas accumulation in the working area at this time, and the gas concentration in the upper corner was low. By adjusting the PSLs and BPDs, the gas accumulation at the working area and upper corner can be effectively prevented during mining and meet the safety production standards. Research on the PSL effect of cooperative extraction, including parameters of high drilling arrangement, buried pipe extraction arrangement, and high extraction lane arrangement, will be carried out in the follow-up study to improve the safety and efficiency of coal and gas co-mining.

## 5. Engineering Practices

In light of the recent abnormal increase in gas concentration in the main mining area, the effect law of PSLs distance on gas gushing out was investigated. Before the 25th day, the PSLs were at most 30 m. The peak gas concentration was in the upper right-hand corner, with an average gas concentration of 0.89%. As the PSLs increased, the angle between the working area and the inlet and outlet lane changed, affected the flow of wind at the working area. It took time for the gas to rise and float and for the airflow to change. As a result, changing the PSLs did not immediately change the gas concentration at the working area and upper corner. The change in gas concentration had hysteresis. The 13th day saw a change as the PSLs gradually increased. On the 17th day, the corner gas concentration gradually decreased, and the PSL was 25 m. As the PSL continued to grow, the rate of gas concentration reduction in the upper corner decreased. The high gas concentration in the extraction area entered the working area earlier. From day 33 onwards, the gas concentration at the working area increased. On the 40th day, the length of the PSL increased to 35 m, and the high concentration of gas in the mining area continued to enter the working area earlier, reaching its peak at this time. After the 45th day, the PSLs decreased, as did the gas concentration at the working area. Figure 16 shows the pattern of change.

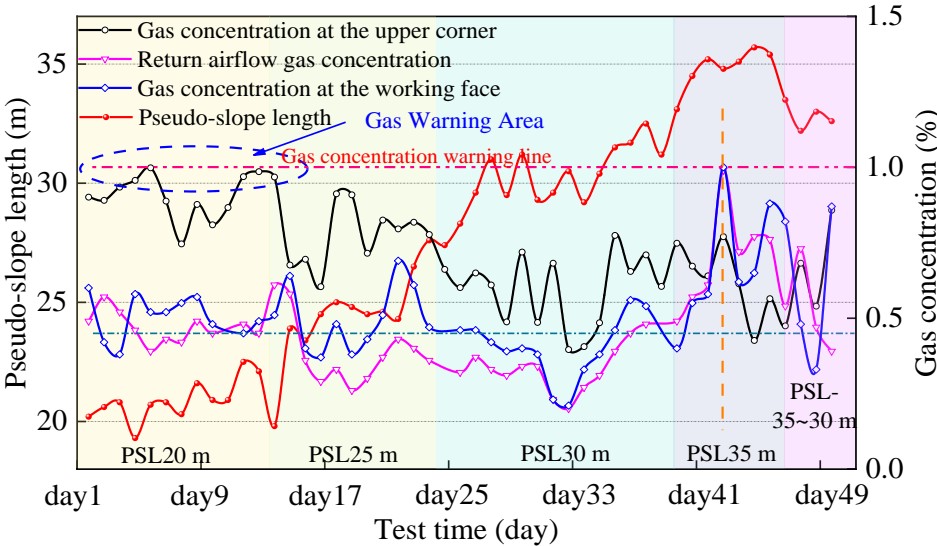

**Figure 16.** Curve of PSLs Distance and Gas Concentration.

The variation pattern shown in Figure 17 was obtained by conducting continuous field observations of gas concentration along the working area. As the PSLs increased, the gas anomaly area gradually shifted from the 122 #–124 # bracket to the incoming wind side. When the PSLs reached 35 m, the gas anomaly zone appeared slowly at the 103 #–108 # frame. When the PSLs gradually decreased after 44 days, the working area's gas anomaly area shifted back to the backwind lane side.

The buried pipe was buried in the extraction area to extract gas from the upper corner, combined with the numerical simulation experiment results to reduce the problem of local gas accumulation. The PSL was controlled on-site to be about 25 m, and two trips of φ219 mm steel pipes were laid along 700 mm of the bottom plate of the return tunnel. Each journey of steel pipe was connected with the main line in the return wind tunnel, and valves were installed on each BPD to control them, respectively. We installed an upward tee in the pipeline every 12 m and started pumping when the first trip of buried pipe entered 12 m into the mining air. When the first buried pipe entered 24 m into the mining area, we disconnected this buried pipe and started pumping the second buried pipe. We continuously extended the connection forward to ensure that the burial depth of

the pumping port was around 20 m, and constantly monitored for 180 days. The buried pipe plan of the working area is shown in Figure 18. As shown in Figure 19, the monitoring results shown that after optimizing the buried depth parameters of the extraction port and the PSLs conditions, the average gas concentration in the upper corner was 0.26%, and the average gas concentration in the return wind tunnel was 0.22%. The maximum gas concentration in the upper corner was 0.46%, and the highest in the return airway was 0.41%. As stipulated in the Coal Mine Safety Regulations, it was far below the warning line of 1% gas concentration in the upper corner, return air tunnel, and tail shaft. By applying the experimental results of numerical simulation to optimize the PSLs of the working area and the BPD in the mining area, the maximum gas concentration in the upper corner and return wind lane can be effectively controlled below 1%. It was practical and feasible for gas prevention and control of the test working area and realizing the efficient production of the tilted coal seam extended release working area.

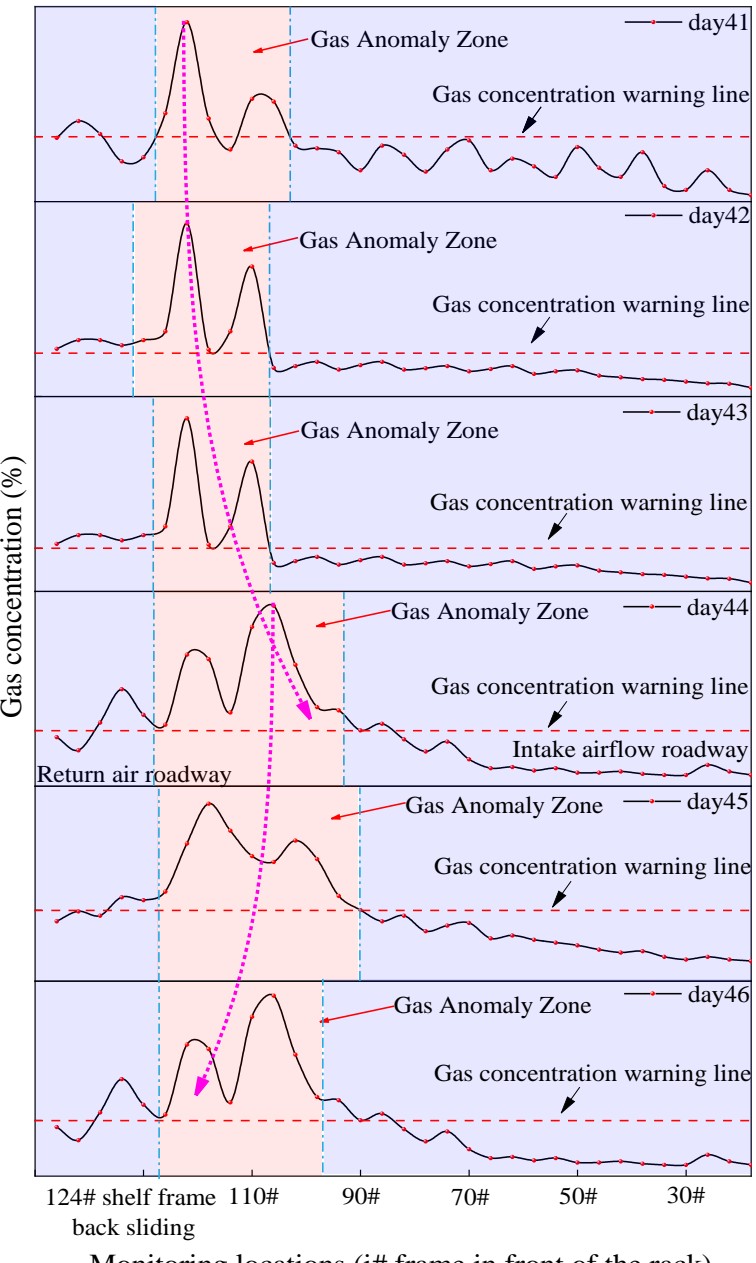

**Figure 17.** Distribution law of gas concentration along working area from day 41 to day 46.

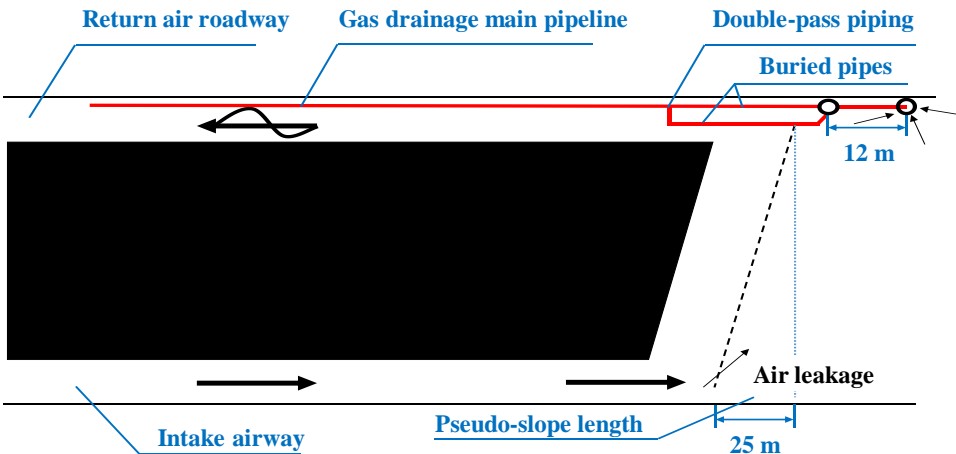

**Figure 18.** Plan view of buried pipe extraction in the extraction area.

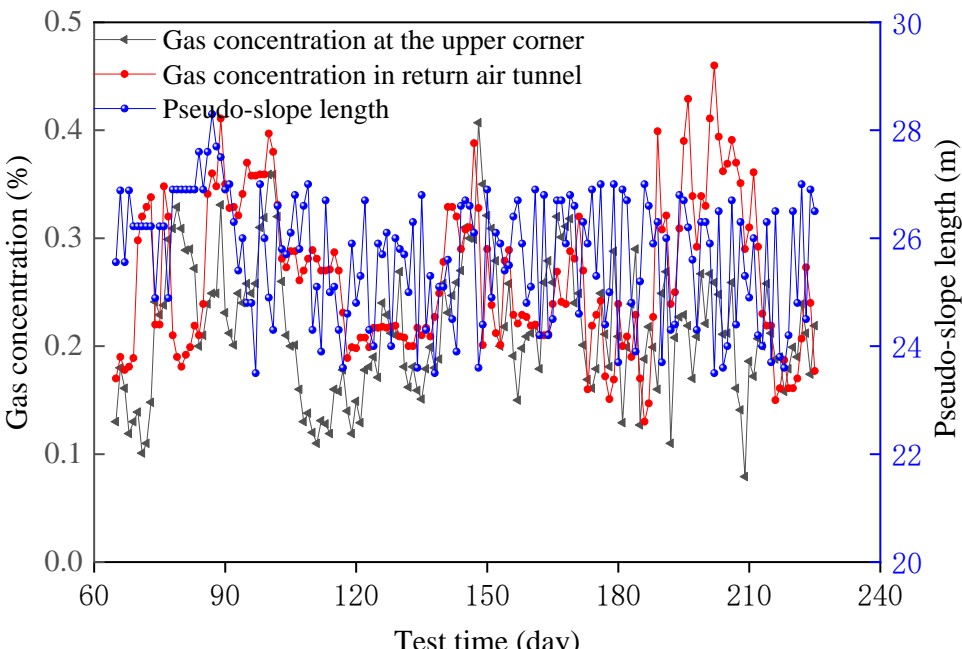

**Figure 19.** Change the pattern of gas concentration in the upper corner and return the air tunnel after optimization.

## 6. Conclusions

Through this study, three conclusions were reached:

(1) The influence law of PSLs on the wind flow in the mine area was obtained by Fluent simulation. With the increased PSL, the air leakage area gradually changed from the upper corner to the middle of the working area and the upper corner. The abnormal gas concentration area shifted to the inlet side. In contrast the gas concentration in the upper corner gradually decreased. The numerical simulation results showed that the optimal PSL was 25 m when there was no extraction measure.

(2) When PSLs and BPD work together, the gas concentration in the corner of the working area can be effectively reduced. When the extraction opening was buried shallowly, the negative pressure formed by extraction would form a low gas concentration zone with air leakage. With the increase in BPD, the effect of BPD on the control of gas concentration in the upper corner gradually decreased. The results show that when PSL was 25 m and BPD was 20 m, it had the best effect on the gas control in the upper corner.

(3)    The change of PSLs and the depth of the extraction port significantly affected the gas concentration distribution on the working area. When the PSLs exceeded 25 m, the measured gas concentration in the field caused a gas anomaly at the 122# frame, and when the PSLs increased to 35 m, the measured gas concentration in the area caused a gas anomaly at the 108# frame. Therefore, the PSLs should be, at most, 25 m. Combined with the numerical simulation results, the gas concentration in the upper corner gradually decreased with the increase in the PSL. Therefore, the best PSL should be 25 m. When the gas concentration in the upper corner was low, no abnormal gas zone appeared in the working area. Based on the numerical simulation results, the maximum gas concentration in the upper corner was 0.46%, with the extraction port of BPD-20 m arranged in the return air tunnel. The numerical simulation was consistent with the engineering practice results. The gas concentration at the working area was effectively controlled by adjusting the PSLs and BPDs to realize the efficient production of the inclined coal seam extended release working area.

**Author Contributions:** Conceptualization, P.Z.; methodology, S.L.; data curation, X.K. and X.A.; writing—original draft preparation, X.A.; investigation, Y.H.; writing—review and editing, J.Y.; funding acquisition, S.J. All authors have read and agreed to the published version of the manuscript.

**Funding:** This research was funded by the National Natural Science Foundation Grants of China (5217-4205 and 5197-4237), the Basic Research on National Natural Science Foundation critical project of China (5173-4007), and the Xinjiang Natural Science Foundation (2019D01B42). The authors are obliged to the Xinjiang Liuhuanggou coal mine for providing field tests supported in this study.

**Institutional Review Board Statement:** Not applicable.

**Informed Consent Statement:** Not applicable.

**Data Availability Statement:** Data available on request due to privacy restrictions.

**Conflicts of Interest:** The authors declare no conflict of interest.

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
