# Peer review of "Study on the Pseudo-Slope Length Effect of Buried Pipe Extraction in Fully Mechanized Caving Area on Gas Migration Law in Goaf"

_sustainability, doi:10.3390/su15086628_

Round 1

Reviewer 1 Report

This paper presents a study on optimizing the pseudo-slope length and optimizing the buried pipe extraction scheme in high gas inclined coal seams, a topic selected for its practical and academic value. The authors adopt the theoretical analysis, numerical simulation and field monitoring to study the optimal pseudo-slope length of the working face and the optimal buried pipe depth on the return side in the mining process of inclined coal seam. The methods are detailed and reliable, and the conclusions are supported. The following comments are potentially useful to improve the quality of the paper, and the revised paper can be accepted.

1. The abstract should be improved to make it concise.

2. Line 63-68: The authors should add some references of recent years.

3. The rock seam column diagram in Figure 1 is not clear, and the specific location of the mined coal seam cannot be seen.

4. The clarity of Fig. 4 should be improved, and the size of the label in the figure is different.

5. "2.2.2 Boundary conditions and parameter setting": Some formulas should be added with corresponding references.

6. What position does the gas concentration of working face in figure 16 refer to?

Author Response

Please see the attached "Response to Reviewer 1 Comments".

Reviewer 2 Report

1.      The manuscript title is suggested to be reconsidered.

2.      The introduction needs to further focus on key scientific questions.

3.      It is suggested to add further comparative analysis data of the experimental results.

4.      It is suggested that the concluding section of the manuscript be further highly summarized.

Author Response

Please see the attached "Response to Reviewer 2 Comments".

Reviewer 3 Report

·         The table in the figure 1 is not readable.

·         Figure ž is no readable

·         The list of literature is very limited to Chinese authors. No scientist in the world has dealt with this topic?

·         In Equation 12 is missing the bracket

·         The paper contains very interesting data, but with respect to references and the geological area, the results are of very limited applicability.

·         The air flow in the coalface and in the gob depends on the pressure distribution in the wind network. The greater the depression in the coalface between intake and return air ways and behind the gob, the more air flows and the more methane is released. In this article is missing this type of information. It is possible to complete to the picture the nodal points and pressure drop?

Author Response

Please see the attached "Response to Reviewer 3 Comments".

Reviewer 4 Report

In this article, the detailed arrangement parameters of buried pipe extraction are proposed based on the actual mining situation at the site of Liuhuanggou coal mine by combining numerical simulation and field inspection, and are successfully applied to the actual site engineering. The structure of the article is reasonable, and the purpose and conclusion are clear, but there are some problems that need author’s careful consideration.

1. Figure 14 needs to be further optimized.

2. How to understand the gas concentration in the article?

3. This article applies Fluent for numerical simulation, the article lacks the relevant description of Fluent calculation model setting, it is suggested to make additional explanation.

4. Some formula variables are used repeatedly in the article, please check and modify them.

5. It is recommended to add 2-3 references of Sustainability journal in the recent 2-5 years.

Author Response

Please see the attached "Response to Reviewer 4 Comments".

Reviewer 5 Report

The article corresponds to the subject of the journal

Well structured. Written clearly and understandably.

Keywords and abstract are consistent with the conclusions

All references are available

Currently, the main mode of transport is pipeline. For the movement of gas through the pipe, you need to create pressure. Gas at a pressure of 7.5 MPa is pumped through pipes with a diameter of up to 1.4 m. As the gas moves away from the gas fields, the pressure decreases, and, consequently, the speed decreases.

The construction and maintenance of the pipeline is costly, but it is worth it. This is the cheapest way to transport gas over short and medium distances.

Aswith any phenomenon, transportation through gas pipelines also has disadvantages:

- At the first stages of pipeline construction, large capital investments are needed. Therefore, construction is economically justified only under the condition of a large and stable gas flow;

- There isa limit on the number of energy grades transported through a single pipeline. The pipeline has an "expiration date";

The route of the pipeline is difficult to change. If new energy consumers appear, then additional investments are needed.

Поэтому тема исследования Study on the Effect of Coupling of Pseudo-Slope Lengths and  Buried Pipe Depths on Gas Concentration in Upper Corner of  Inclined Thick Coal Seam in Xinjiang является актуальной не только для Китая,  но для других стран.

The novelty of the studies performed is the conduct of modeling.

To study the law of gas transportation in the mining area, Fluent numerical simulation  software was applied to study the gas distribution and flow field in the mining area under U-type  ventilation mode from the pseudo-slope lengths (PSLs) of 20m, 25m, 30m, and 35m respectively; and on this basis, different buried pipe extraction depths under different PSLS were studied .

The study results provide essential theoretical guidance for the prevention and control of gas disasters in the upper corner of inclined thick coal seam comprehensive mining.

I would like to test the theoretical results in practice.

Comments.

Format the entire article. 

It is not clear how the simulation results are consistent with other authors or practice

Author Response

Please see the attached "Response to Reviewer 5 Comments".

Round 2

Reviewer 2 Report

The quality of the revised manuscript has been significantly improved. It is recommended to accept it!

Reviewer 3 Report

All comments have been corrected and the text added.